# Construction of Ag/Ag_2_S/CdS Heterostructures through a Facile Two-Step Wet Chemical Process for Efficient Photocatalytic Hydrogen Production

**DOI:** 10.3390/nano13121815

**Published:** 2023-06-07

**Authors:** Yu-Cheng Chang, Ying-Ru Lin

**Affiliations:** 1Department of Materials Science and Engineering, Feng Chia University, Taichung 407102, Taiwan; d0455077@gmail.com; 2Department of Materials Science and Engineering, National Yang Ming Chiao Tung University, Hsinchu 30010, Taiwan

**Keywords:** wet chemical, Ag/Ag_2_S/CdS heterostructures, photocatalytic hydrogen evolution, surface plasma resonance, seawater

## Abstract

We have demonstrated a two-step wet chemical approach for synthesizing ternary Ag/Ag_2_S/CdS heterostructures for efficient photocatalytic hydrogen evolution. The CdS precursor concentrations and reaction temperatures are crucial in determining the efficiency of photocatalytic water splitting under visible light excitation. In addition, the effect of operational parameters (such as the pH value, sacrificial reagents, reusability, water bases, and light sources) on the photocatalytic hydrogen production of Ag/Ag_2_S/CdS heterostructures was investigated. As a result, Ag/Ag_2_S/CdS heterostructures exhibited a 3.1-fold enhancement in photocatalytic activities compared to bare CdS nanoparticles. Furthermore, the combination of Ag, Ag_2_S, and CdS can significantly enhance light absorption and facilitate the separation and transport of photogenerated carriers through the surface plasma resonance (SPR) effect. Furthermore, the Ag/Ag_2_S/CdS heterostructures in seawater exhibited a pH value approximately 2.09 times higher than in de-ionized water without an adjusted pH value under visible light excitation. The ternary Ag/Ag_2_S/CdS heterostructures provide new potential for designing efficient and stable photocatalysts for photocatalytic hydrogen evolution.

## 1. Introduction

Hydrogen is a clean alternative to fossil fuels with a high energy content and minimal environmental pollution [1,2]. Converting solar energy to hydrogen via photocatalysts is a promising hydrogen production technology due to its low price and easy availability [3,4]. However, photocatalysts have low hydrogen production efficiency, a complex preparation process, and a high price, significantly limiting their applications [5,6]. Recently, different strategies have been used to improve existing photocatalysts to overcome obstacles in photocatalytic hydrogen production, such as utilizing noble metals, constructing novel heterojunctions, optimizing electronic structures, defect engineering, and facet engineering [7,8,9]. Photocatalytic hydrogen evolution depends on three critical factors: efficient light harvesting, effective charge separation/transport, and proficient surface chemical reactions [10,11]. Moreover, since the solar spectrum contains a significant portion (43%) of visible light, the development of visible-light-responsive photocatalysts is imperative for advancing the potential of industrial applications [12,13,14].

Cadmium sulfide (CdS) is an n-type semiconductor material widely used as the most promising visible light photocatalyst due to its middle band gap of 2.0–2.4 eV, suitable photo-redox potential, and efficient electron–hole pair generation [15,16,17,18]. However, the rapid recombination of charge carriers and severe photocorrosion caused by S^2−^ oxidation lead to low quantum efficiency, limiting the application of photocatalytic hydrogen production [19,20,21]. The noble metal is generally considered a co-catalyst, which can be crucial in lowering the activation barrier for surface redox reactions [22,23]. Additionally, the surface plasmon resonance (SPR) of noble metals (such as Au or Ag) can enhance the efficiency of visible light absorption and promote the transfer of high-energy electrons to the conduction band (CB) through hot electron injection [24,25]. SPR is the coherent oscillation of electrons caused by resonant photons, which can generate enhanced electromagnetic fields around metal nanostructures to confine light at the interface of metal and semiconductors and increase light absorption efficiency [26]. Moreover, noble metals can create electron-trapping sites for low-energy interfacial carriers, substantially enhancing photocatalytic activity [24,27]. Therefore, Ag has gained significant attention from researchers as a promising alternative to Au in photocatalytic hydrogen production. This result is primarily attributed to its stronger surface plasmon resonance (SPR), abundance, and cost-effectiveness [28]. In recent times, Ag-CdS heterostructures have become popular as efficient photocatalysts for water splitting, such as CdS/Ag/TiO_2_ [24,29], Ti_3_CN@TiO_2_/CdS [30], and ZnO/Ag/CdS [31]. This enhanced photocatalytic activity is attributed to the different material heterojunctions, which can enhance the separation of photogenerated charge carriers, extended light absorption, and enriched active sites [32].

Moreover, most of the water used in photocatalytic water splitting for hydrogen generation is either de-ionized or pure water [33]. However, de-ionized and pure water require freshwater purification, necessitating additional resources [34]. Limited research is available on the photocatalytic decomposition of seawater to produce hydrogen [35]. Considering the growing scarcity of freshwater resources, utilizing seawater directly for photocatalytic decomposition to generate hydrogen could help minimize freshwater consumption [36]. Thus, this study aims to produce hydrogen through the photocatalytic decomposition of seawater using a ternary Ag/Ag_2_S/CdS heterostructure.

Herein, novel Ag/Ag_2_S/CdS heterostructures were successfully synthesized for efficient photocatalytic hydrogen evolution through a facile two-step wet chemical method. Furthermore, Ag/Ag_2_S/CdS heterostructures demonstrate the highest photocatalytic hydrogen production activity, exhibiting a 3.1-fold increase compared to pure CdS nanoparticles under visible light excitation. Remarkably, the Ag/Ag_2_S/CdS heterostructures can also be used as a high-performance photocatalyst for seawater splitting under visible light excitation.

## 2. Materials and Methods

### 2.1. Synthesis of Ag Nanostructures

A modified literature process synthesized silver nanostructures through a simple wet-chemical route [37]. In a typical synthesis of Ag nanostructures, 30 mL of ethylene glycol (Alfa Aesar, Haverhill, MA, USA, 99%) was heated in a round-bottomed flask at 160 °C for 30 min. Then, 240 μL of a 4 mM copper(II) chloride dihydrate (Alfa Aesar, 99+%) solution in ethylene glycol (Alfa Aesar, 99%), 9 mL of a 440 mM poly(N-vinylpyrrolidone) (PVP, Acros Organic, MW, Veneto, Italy: 50,000) solution in ethylene glycol, and 9 mL of a 94 mM silver nitrate (Alfa Aesar, 99+%) solution in ethylene glycol were, respectively, injected into the heated ethylene glycol at 160 °C for 90 min. Finally, the Ag nanostructure solution was cooled down to room temperature.

### 2.2. Synthesis of Ag/Ag_2_S/CdS Heterostructures

A simple wet-chemical route was also used to synthesize Ag/Ag_2_S/CdS heterostructures on the Ag nanostructures. In a typical synthesis of Ag/Ag_2_S/CdS heterostructures, 160 mL of ethylene glycol was dissolved in the different concentrations of CdS precursors (cadmium acetate dihydrate (Alfa Aesar, 98%) and thioacetamide (TAA, Alfa Aesar, 98%)) and then added to 40 mL of Ag nanostructures under magnetic stirring (400 rpm) for 30 min. Next, the solution was heated at different reaction temperatures (120, 140, 160, and 180 °C) for 2 h. Finally, the Ag/Ag_2_S/CdS heterostructures were washed with ethanol to remove the excess precursors, collected, and centrifuged at 3000× *g* rpm for 15 min.

### 2.3. Characterization

The Ag/Ag_2_S/CdS heterostructures were characterized using various techniques such as X-ray diffraction (XRD) with a Bruker D2 phaser system (Billerica, MA, USA), field-emission scanning electron microscopy (FESEM) with a JEOL JSM 6700F microscope (Tokyo, Japan), field-emission transmission electron microscopy (FETEM) with a JEOL 2100F microscope (Tokyo, Japan), X-ray photoelectron spectroscopy (XPS) with a ULVAC-PHI PHI 5000 Versaprobe II system (Chigasaki, Japan), photoluminescence spectroscopy (PL) with an MRI532S instrument from Protrustech (Tainan, Taiwan), and UV–vis spectrophotometry with a U-2900 instrument from Hitachi (Tokyo, Japan). These techniques were employed to study the crystal structure, morphology, microstructures, chemical composition, and optical properties of the Ag/Ag_2_S/CdS heterostructures.

### 2.4. Photocatalytic Hydrogen Production Experiment

An experiment on photocatalytic hydrogen evolution was conducted using the PCX50B Discover photocatalytic reaction system from Beijing Perfect Light Technology (Beijing, China). For the experiment, 25 mg of as-synthesized photocatalysts were mixed with 50 mL of different water bases, including de-ionized water, reverse osmosis water, tap water, and seawater. Additionally, 0.1 M of various sacrificial agents such as sodium sulfide, sodium sulfate, methanol, and methanoic acid were added to the water bases. The mixture was placed in a 60 mL quartz tube, and Ar gas was introduced for 30 min to remove air. The quartz tube was then sealed with a rubber stopper, and the photocatalytic reaction was irradiated for 3 h using a 5 W blue LED as a visible light source. Subsequently, the hydrogen produced was quantified using gas chromatography (GC, Shimadzu GC-2014, Kyoto, Japan) equipped with a packed column (MS-5A, 60/80 mesh) and a thermal conductivity detector (TCD).

## 3. Results and Discussion

A powder XRD analysis measured the crystallographic structure and phase purity of as-synthesized Ag/Ag_2_S/CdS heterostructures. XRD patterns of the as-synthesized Ag/Ag_2_S/CdS heterostructures with the different concentrations of CdS precursors are shown in Figure 1a–e. The concentrations of CdS precursors are 2.5, 5, 10, 20, and 25 mM, respectively. For Ag, the four diffraction peaks at 38.3°, 44.5°, 64.7°, and 77.7°correspond to the (111), (200), (220), and (311) planes of typical cubic Ag (JCPDS card No. 87–0719), respectively. For Ag_2_S, the seven diffraction peaks at 28.9°, 31.5°, 33.6°, 34.4°, 34.7°, 37.7°, and 40.9° correspond to the (111), (−112), (120), (−121), (022), (−103), and (031) planes of typical monoclinic Ag_2_S (JCPDS card No. 89–3840), respectively. For CdS, the six diffraction peaks at 24.9°, 26.6°, 28.3°, 36.8, 43.9°, and 52.1° correspond to the (100), (002), (101), (102), (110), and (112) planes of typical hexagonal CdS (JCPDS card No. 80-0006), respectively. The diffraction peaks’ intensity for CdS gradually increases with the increased concentration of CdS precursors. In addition, diffraction peaks’ intensity for Ag reveals the reverse tendency. This observation demonstrates that Ag_2_S and CdS have been successfully loaded onto Ag nanostructures to form Ag/Ag_2_S/CdS heterostructures.

XRD patterns of the as-synthesized Ag/Ag_2_S/CdS heterostructures with the different reaction temperatures are shown in Figure 2a–d. The reaction temperatures are 120, 140, 160, and 180 °C. The diffraction peaks of the synthesized Ag/Ag_2_S/CdS heterostructures are well consistent with the Ag (JCPDS card No. 87-0719), Ag_2_S (JCPDS card No. 89-3840), and CdS (JCPDS card No. 80-0006), respectively. In addition, no impurities were detected in the XRD results, demonstrating the phase purity of the as-prepared Ag/Ag_2_S/CdS heterostructures under the different reaction temperatures. The intensity of diffraction peaks for Ag decreases gradually as the reaction temperature increases. However, when the reaction temperature exceeds 180 °C, CdS nanoparticles tend to self-aggregate instead of reacting with Ag nanowires to form Ag/Ag_2_S/CdS heterostructures [22]. This phenomenon leads to an increase in the intensity of the Ag diffraction peaks and a decrease in the Ag_2_S diffraction peaks. Moreover, the average sizes of Ag_2_S can be determined using Scherrer’s formula based on X-ray line broadening: D = 0.9λ/βcosθ. Here, D represents the crystallite size, λ (equal to 1.54096 Å) denotes the X-ray wavelength, β represents the full width at half-maximum (FWHM), and θ refers to the diffraction angle. The average crystalline sizes of the Ag_2_S are calculated using the (−103) peak at the different reaction temperatures. The average crystalline sizes of Ag_2_S are 2.14 (120 °C), 2.39 (140 °C), 2.66 (160 °C), and 1.86 nm (180 °C), respectively. The crystalline sizes of Ag_2_S exhibit an opposite trend compared to the intensity of the Ag diffraction peaks. This finding confirms that CdS nanoparticles tend to self-aggregate instead of reacting with Ag nanowires to produce Ag/Ag_2_S/CdS heterostructures at a reaction temperature of 180 °C.

Figure 3a shows the FETEM image of an Ag nanowire with a smooth surface and a diameter of 89 nm. Figure 3b reveals an Ag/Ag_2_S/CdS heterostructure with a rough surface. This result is ascribed to an Ag nanowire decorated with Ag_2_S and CdS nanostructures. The HRTEM image in Figure 3c displays an Ag/Ag_2_S/CdS heterostructure. The interlayer spacings of the heterostructure are measured at 0.236 nm, 0.268 nm, and 0.355 nm, corresponding to the d-spacing of the (111) lattice plane of cubic Ag (JCPDS card No. 87-0719), the (−121) lattice plane of monoclinic Ag_2_S (JCPDS card No. 89-3840), and the (100) lattice plane of hexagonal CdS (JCPDS card No. 80-0006), respectively. Moreover, the composition of an Ag/Ag_2_S/CdS heterostructure can be confirmed using EDS mapping images (depicted in Figure 3d) that detect signals of Ag, Cd, and S. This result indicates that there are no additional impurities present in the Ag/Ag_2_S/CdS heterostructure, thereby reinforcing its compositional structure.

Figure 4a depicts the FETEM image of CdS nanoparticles comprising numerous small grains under a CdS precursor concentration and reaction temperature of 20 mM and 160 °C, respectively. The selected area electron diffraction (SAED) pattern in Figure 4b exhibited polycrystalline diffraction rings with concentric rings indexed to the hexagonal CdS (JCPDS card No. 80-0006). This finding was consistent with the XRD result of the above Ag/Ag_2_S/CdS heterostructures. Figure 4c shows the HRTEM image of CdS nanoparticles, displaying two lattice spacings of 0.355 and 0.205 nm corresponding to the (100) and (110) crystal planes of hexagonal CdS (JCPDS card No. 80-0006). Figure 4d demonstrates the EDS mapping of Cd and S elements in the CdS nanoparticles, which are uniformly distributed throughout the structure.

The overall XPS survey spectrum (Figure 5a) of the Ag/Ag_2_S/CdS heterostructures indicates the existence of Ag, O, Cd, S, and C elements. The presence of the C 1s can be ascribed to the pump oil in the vacuum system of the XPS equipment or the organic layer (PVP) coated on the Ag nanostructures. Figure 5b shows that the high-resolution XPS spectrum of Ag 3d shows two peaks at 367.2 eV and 373.2 eV, which are assigned to Ag 3d_5/2_ and Ag 3d_3/2_, respectively, indicating the presence of Ag^+^ [33,34]. Additionally, the high-resolution XPS spectrum of O 1S (Figure 5c) at 532.2 eV is attributed to the carboxyl (C=O) oxygen in the PVP [35]. The doublet peaks of Cd 3d in the high-resolution XPS spectrum (Figure 5d) at 404.6 and 411.4 eV correspond to Cd 3d_5/2_ and Cd 3d_3/2_, respectively, indicating the +2 oxidation state of Cd in CdS [36]. Moreover, the high-resolution XPS spectrum (Figure 5e) of S 2p shows two peaks at 161.0 and 162.2 eV, which correspond to S 2p_3/2_ and S 2p_1/2_ of S^2−^ in CdS or Ag_2_S [37,38]. These XPS results indicate the coexistence of Ag_2_S and CdS in Ag/Ag_2_S/CdS heterostructures.

The effectiveness of the synthesized Ag/Ag_2_S/CdS heterostructures as a photocatalyst was assessed by measuring the hydrogen evolution rate (HER) in a solution of 50 mL de-ionized water with 0.1 M sodium sulfide (Na_2_S) acting as a scavenger at pH 12 under visible light excitation. Figure 6a displays the average HER of Ag/Ag_2_S/CdS heterostructures produced at various concentrations of CdS precursors at 160 °C for 2 h. The average HER values for Ag/Ag_2_S/CdS heterostructures were 78.0 (2.5 mM), 94.4 (5 mM), 354.8 (10 mM), 2531.4 (20 mM), and 540.3 μmolh^−1^g^−1^L^−1^ (25 mM), respectively. The average HER of Ag/Ag_2_S/CdS heterostructures increased gradually with an increase in the concentration of CdS precursor. However, the average HER of Ag/Ag_2_S/CdS heterostructures exhibited a significant decline with the concentration of CdS precursor exceeding 20 mM. This result may be attributed to higher reaction concentrations of CdS precursor, which may lead to excessive generation of CdS nanoparticles and a decrease in the transfer efficiency of electron–hole pairs, thereby inhibiting the photocatalytic hydrogen production efficiency. Figure 6b reveals the average HER of Ag/Ag_2_S/CdS heterostructures (20 mM CdS precursor) produced at various reaction temperatures for 2 h. The average HER values for Ag/Ag_2_S/CdS heterostructures were 611.8 (120 °C), 995.4 (140 °C), 2531.4 (160 °C), and 344.8 μmolh^−1^g^−1^L^−1^ (180 °C), respectively. The average HER of Ag/Ag_2_S/CdS heterostructures increased gradually with the reaction temperatures. However, at reaction temperatures above 160 °C, the average HER of Ag/Ag_2_S/CdS heterostructures experienced a notable decrease. This result is attributed to the faster self-nucleation process that generates CdS nanoparticles, which become more prominent at higher temperatures. Consequently, more CdS nanoparticles are produced per unit mass of the photocatalyst, resulting in a more noticeable photocorrosion phenomenon and decreased photocatalytic hydrogen production.

Sacrificial reagents are commonly utilized in photocatalytic water splitting to improve the efficiency of oxidation reactions in aqueous media due to the inefficiency of pure water oxidation [38]. Figure 6c shows the average HER of Ag/Ag_2_S/CdS heterostructures (20 mM CdS precursors and 160 °C) under visible light excitation with four types of sacrificial reagents: methanol (CH_3_OH), methanoic acid (HCOOH), sodium sulfide (Na_2_S), and sodium sulfate (Na_2_SO_4_). The HER values of Ag/Ag_2_S/CdS heterostructures were 0 (CH_3_OH), 62.1 (HCOOH), 2531.4 (Na_2_S), and 71.5 μmolh^−1^g^−1^L^−1^ (Na_2_SO_4_), respectively. During photocatalytic hydrogen production, a sacrificial agent serves as an electron donor to provide electrons for proton reduction and as a hole scavenger to prevent electron–hole pair recombination, thereby improving photocatalytic efficiency. Therefore, using Na_2_S as a sacrificial reagent can improve the reduction/oxidation reaction, reduce CdS material photocorrosion, and enhance its photocatalytic hydrogen production capacity [39,40].

The optimal pH level for photocatalytic hydrogen production is primarily determined by the characteristics of the sacrificial agent and its ability to adsorb onto the surface of the photocatalyst [41]. Figure 6d depicts the effect of pH values on the photocatalytic efficiency of Ag/Ag_2_S/CdS heterostructures (20 mM CdS precursors and 160 °C). The average HER values of Ag/Ag_2_S/CdS heterostructures were 16.7 (pH = 3), 1138.4 (pH = 6), 2531.4 (pH = 9), and 1100.1 μmolh^−1^g^−1^L^−1^ (pH = 12), respectively. An increase in the pH value from 3 to 9 was observed to significantly improve the photocatalytic hydrogen production efficiency. This phenomenon can be attributed to the enhanced dissociation of HS^−^ and S^2−^ with increasing pH values [37]. Excessive hydroxide ions can reduce photocatalytic hydrogen production efficiency due to the reaction of photogenerated hydrogen ions with hydroxide ions to form water [38]. Thus, the Ag/Ag_2_S/CdS heterostructures exhibited the highest photocatalytic hydrogen production efficiency at pH 9.

Figure 7a presents the average HER of various photocatalysts under visible light excitation, including Ag nanowires, Ag/Ag_2_S/CdS heterostructures (20 mM CdS precursors and 160 °C), CdS nanoparticles (20 mM CdS precursors), and commercial TiO_2_ nanopowders. The photocatalytic performance of the different catalysts was measured in terms of HER values, which were 0 (Ag nanowires), 2531.4 (Ag/Ag_2_S/CdS heterostructures), 804.2 (CdS nanoparticles), and 0 μmolh^−1^g^−1^L^−1^ (TiO_2_ nanopowders). The Ag/Ag_2_S/CdS heterostructures exhibited the highest photocatalytic hydrogen production, about 3.1 times higher than CdS nanoparticles. In addition, the formation of Ag/Ag_2_S/CdS heterostructures through the reaction of Ag nanowires and CdS precursors was found to be more favorable for the transfer of electron–hole pairs, leading to improved efficiency of photocatalytic water splitting for hydrogen generation. Furthermore, the Ag/Ag_2_S/CdS heterostructures exhibit higher HER values compared to other reported studies, including flower-like ZnO/Au/CdS nanorods [42], ZnO/ZnS/CdS heterostructures [43], Ag-based g-C_3_N_4_ composites [44], Cu_2_O/CuS/ZnS nanocomposite [38], and CdS/Pt-N-TiO_2_ nanocatalysts [45].

The stability and reusability of Ag/Ag_2_S/CdS heterostructures were also investigated to evaluate their practical applications. Three consecutive experiments were conducted to investigate the stability of the Ag/Ag_2_S/CdS heterostructures, and the results showed no significant deactivation of the catalysts, with the H_2_ production rate remaining consistent over the three cycles, as shown in Figure 7b. Moreover, this result proves the excellent stability and reusability of Ag/Ag_2_S/CdS heterostructures, suggesting their potential for broader and diverse applications in various fields.

The PL spectra assessed the charge transfer and recombination characteristics of light-excited Ag/Ag_2_S/CdS heterostructures. Due to the high intensity of the PL peak, photoinduced charges experienced a fast recombination rate, leading to decreased photocatalytic performance [46]. Figure 8a reveals the PL spectra of CdS nanoparticles and Ag/Ag_2_S/CdS heterostructures. The CdS nanoparticles showed a stronger visible emission at approximately 569.5 nm (2.18 eV), which was attributed to the electron–hole pair recombination of near-band-edge emission (NBE) of CdS [47,48]. In addition, the intensity of the PL peak in CdS nanoparticles was higher, indicating a higher rate of recombination of photogenerated electrons and holes. On the other hand, the PL spectrum of Ag/Ag_2_S/CdS heterostructures was significantly lower, indicating an increased separation rate of electron–hole pairs. As a result, there was an improvement in the material’s photocatalytic activities. Figure 8b shows the UV–vis absorption spectra of CdS nanoparticles and Ag/Ag_2_S/CdS heterostructures. The Ag/Ag_2_S/CdS heterostructures exhibited greater absorption in the 400–800 nm range compared to CdS nanoparticles, indicating that the addition of Ag and Ag_2_S significantly enhanced the light absorption capability of CdS.

Generating hydrogen gas by splitting seawater would be a more practical approach to conserving freshwater for agricultural, industrial, and human purposes since 93% of Earth’s liquid water is in the oceans [49,50]. The versatility of Ag/Ag_2_S/CdS heterostructures can be showcased by their ability to generate hydrogen through photocatalysis, utilizing different water sources. Figure 9a illustrates the dispersion of Ag/Ag_2_S/CdS heterostructures in 50 mL of various water sources, such as de-ionized water (DI), reverse osmosis water (RO), tap water (TW), and seawater (SW), in the presence of 0.1 M Na_2_S as a scavenger with a pH value of 9 under visible light excitation. The average HER values of Ag/Ag_2_S/CdS heterostructures were 2531.4 (DI), 2463.3 (RO), 2302.1 (TW), and 902.1 (SW) µmolh^−1^g^−1^ L^−1^. As the complexity of the water matrix increases, the efficiency of photocatalytic water splitting tends to decrease gradually. Therefore, to avoid the excessive interference of the water matrix during the pH adjustment process, this study further explored the influence of the photocatalytic decomposition of water from the same water matrix to produce hydrogen under an unadjusted pH value, as shown in Figure 9b. As a result, the average HER values of Ag/Ag_2_S/CdS heterostructures were 905.8 (DI, pH = 13), 1273.4 (RO, pH = 12.9), 1207.8 (TW, pH = 12.9), and 1893.1 (SW, pH = 11.6) µmolh^−1^g^−1^ L^−1^. The Ag/Ag_2_S/CdS heterostructures of SW exhibited a pH value about 2.09 times higher than that of DI under an unadjusted pH value. The efficiency of photocatalytic hydrogen production through water splitting gradually increases with the solution’s pH value approaching 9. Additionally, the results indicate that Ag/Ag_2_S/CdS heterostructures exhibit outstanding photocatalytic performance for hydrogen generation, even in highly complex water matrices, without needing pH value adjustments.

Figure 10 illustrates the mechanism of photocatalytic water splitting of Ag/Ag_2_S/CdS heterostructures, as indicated by the above results. The Ag work function is −4.26 eV relative to the vacuum energy level (equivalent to −0.24 V vs. NHE) [51,52]. The conduction band (CB) positions of CdS and Ag_2_S are −0.55 eV and −0.27 eV, while their valence band (VB) positions are 1.63 eV and 1.33 eV, respectively [53,54,55,56]. Under visible light excitation, photogenerated electrons are excited from the VB to the CB of CdS and Ag_2_S. Under visible light excitation, Ag can also generate electrons through surface plasmon resonance [57]. As a result, the photogenerated electrons in the CB of CdS can be transferred to the CB of Ag_2_S. Additionally, the presence of Ag with Ag_2_S can further facilitate the transfer of electrons from the CB of Ag_2_S to Ag, as a Schottky barrier is formed at the Ag_2_S and Ag interface [58]. Ag nanoparticles function as electron sinks, capturing electrons and reducing H^+^ to H_2_, while the photogenerated holes of CdS can either oxidize water to oxygen or hydrogen ions [33]. This photocatalytic process facilitates the separation of photogenerated charge carriers and promotes their efficient hydrogen production. Furthermore, Ag/Ag_2_S/CdS heterostructures significantly enhance light-harvesting capacity, thus improving the efficiency of photocatalytic hydrogen production.

## 4. Conclusions

In the present study, a facile two-step wet chemical method can be used to synthesize ternary Ag/Ag_2_S/CdS heterostructures, which have proven highly effective in photocatalytic hydrogen evolution. The efficiency of photocatalytic water splitting under visible light excitation was found to be dependent on the reaction temperatures and concentrations of the CdS precursor. We also investigated the impact of operational parameters, such as pH value, sacrificial reagents, reusability, water bases, and light sources, on the photocatalytic hydrogen production of Ag/Ag_2_S/CdS heterostructures. Ag/Ag_2_S/CdS heterostructures exhibited a 3.1-fold increase in photocatalytic activity compared to CdS nanoparticles. Furthermore, the combination of Ag, Ag_2_S, and CdS significantly enhances light absorption and facilitates the separation and transport of photogenerated carriers through the SPR effect. In addition, the Ag/Ag_2_S/CdS heterostructures of seawater exhibited a pH value about 2.09 times higher than that of de-ionized water without adjusted pH values. These findings demonstrate the potential of ternary Ag/Ag_2_S/CdS heterostructures as efficient and stable photocatalysts for photocatalytic hydrogen evolution.

## Figures and Tables

**Figure 1 nanomaterials-13-01815-f001:**
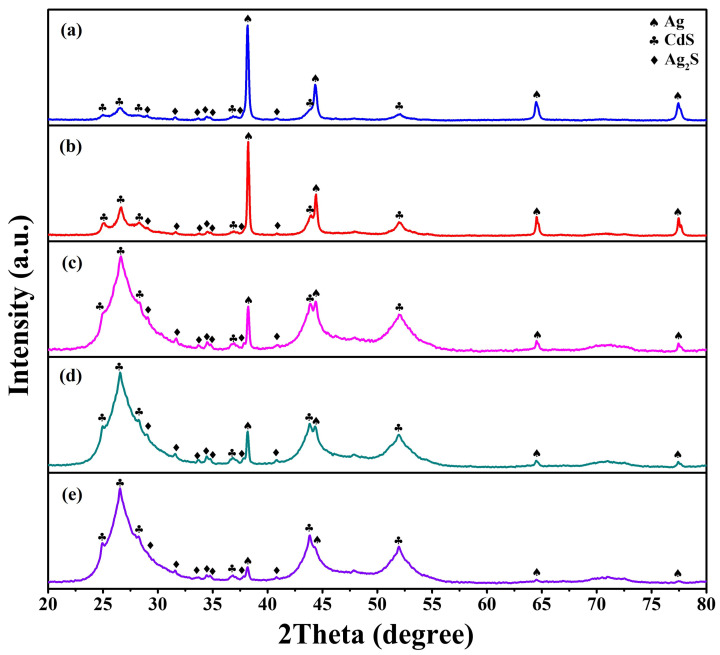
The XRD patterns of Ag/Ag_2_S/CdS heterostructures grown at different concentrations of CdS precursors. The concentrations of CdS precursors were (**a**) 2.5, (**b**) 5, (**c**) 10, (**d**) 20, and (**e**) 25 mM.

**Figure 2 nanomaterials-13-01815-f002:**
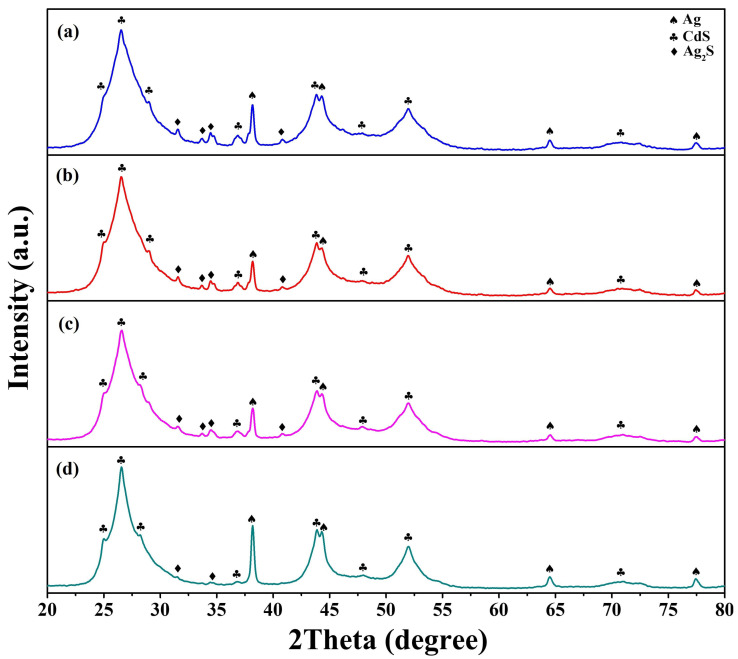
The XRD patterns of Ag/Ag_2_S/CdS heterostructures grown at different reaction temperatures. The reaction temperatures were (**a**) 120, (**b**) 140, (**c**) 160, and (**d**) 180 °C.

**Figure 3 nanomaterials-13-01815-f003:**
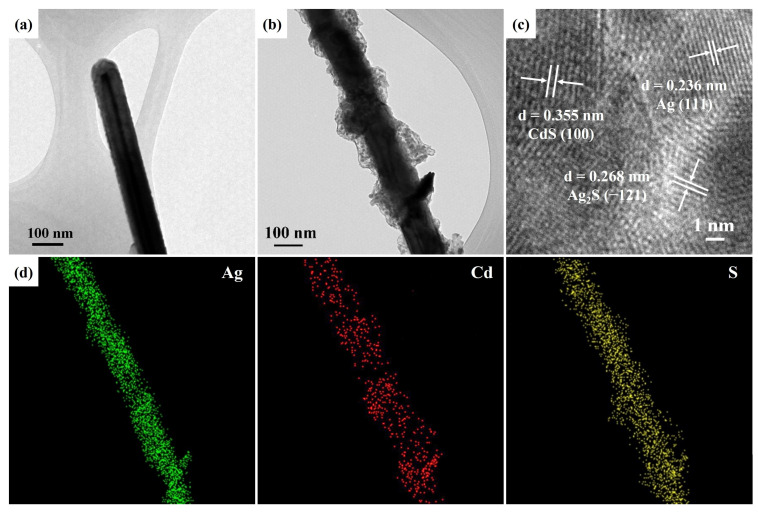
(**a**) FETEM image of an Ag nanowire. (**b**) FETEM, (**c**) HRTEM, and (**d**) EDS mapping images of an Ag/Ag_2_S/CdS heterostructure grown at a CdS precursor concentration and reaction temperature of 20 mM and 160 °C, respectively.

**Figure 4 nanomaterials-13-01815-f004:**
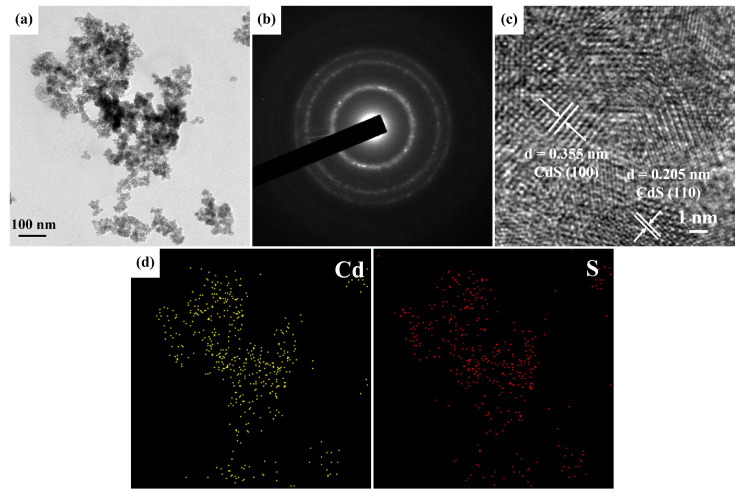
(**a**) FETEM, (**b**) SAED pattern, (**c**) HRTEM, and (**d**) EDS mapping images of CdS nanoparticles grown at a CdS precursor concentration and reaction temperature of 20 mM and 160 °C, respectively.

**Figure 5 nanomaterials-13-01815-f005:**
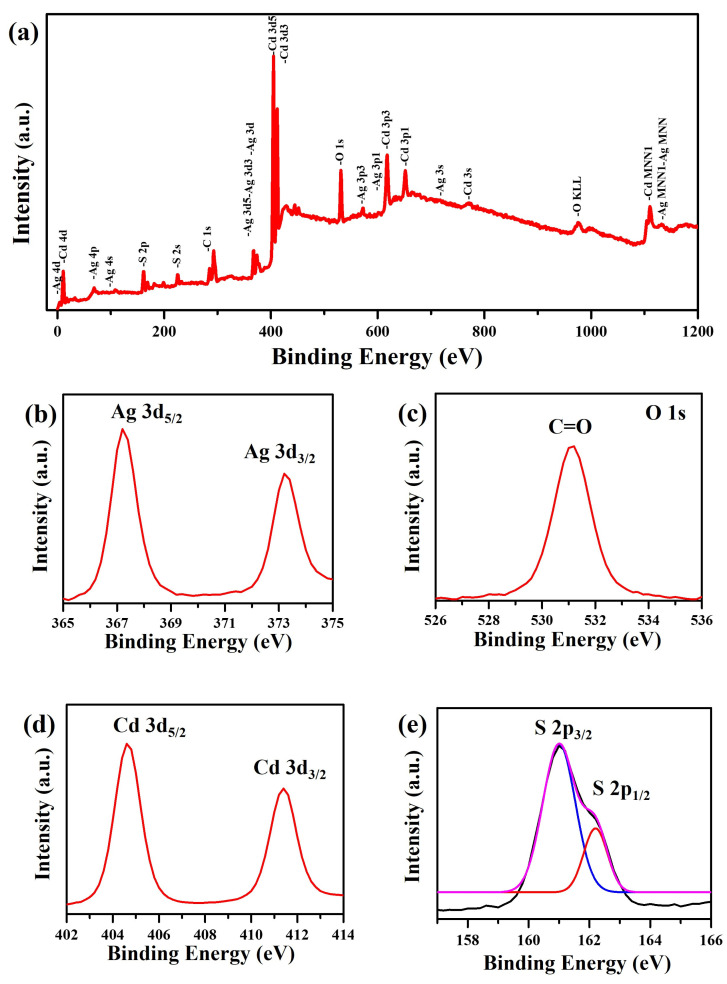
XPS spectra of the Ag/Ag_2_S/CdS heterostructures (20 mM CdS precursor and 160 °C): (**a**) survey spectrum, (**b**) Ag 3d, (**c**) O 1s, (**d**) Cd 3d, and (**e**) S 2p.

**Figure 6 nanomaterials-13-01815-f006:**
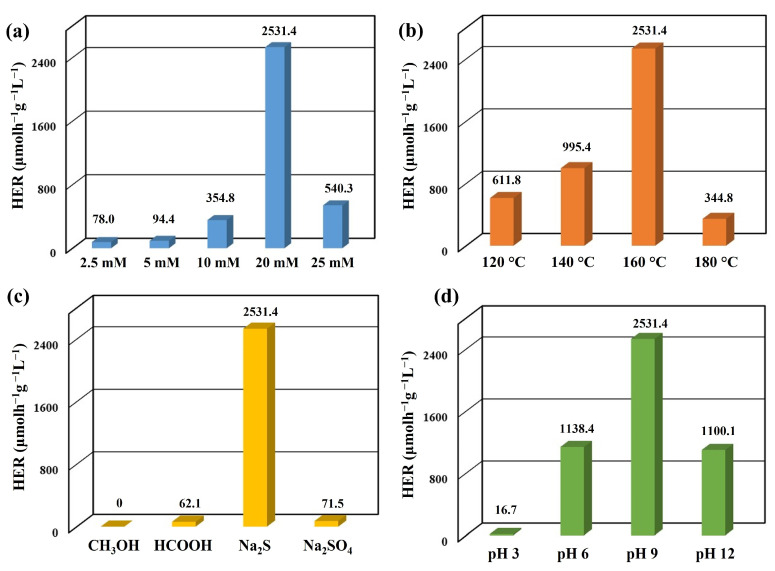
The average HER of Ag/Ag_2_S/CdS heterostructures synthesized at different (**a**) CdS precursor concentrations and (**b**) reaction temperatures. The average HER of Ag/Ag_2_S/CdS heterostructures synthesized at different (**c**) sacrificial reagents and (**d**) pH values.

**Figure 7 nanomaterials-13-01815-f007:**
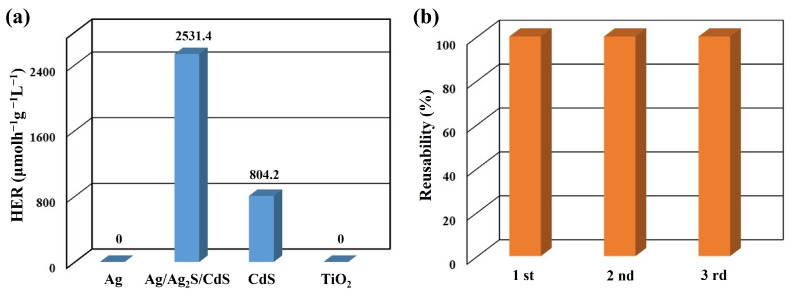
(**a**) The average HER of different photocatalysts. (**b**) Reusability test of Ag/Ag_2_S/CdS heterostructures for three cycles.

**Figure 8 nanomaterials-13-01815-f008:**
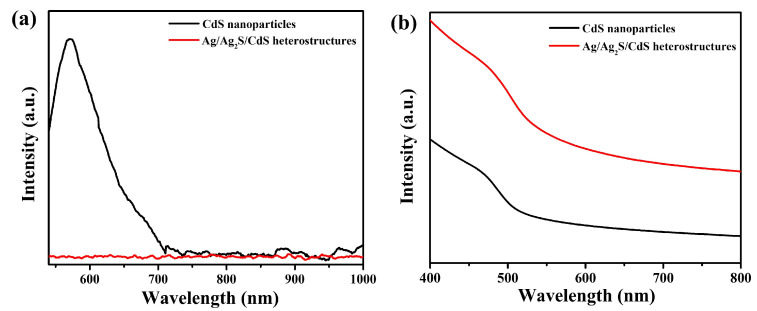
(**a**) PL spectra and (**b**) UV–vis absorption spectra of the CdS nanoparticles and Ag/Ag_2_S/CdS heterostructures.

**Figure 9 nanomaterials-13-01815-f009:**
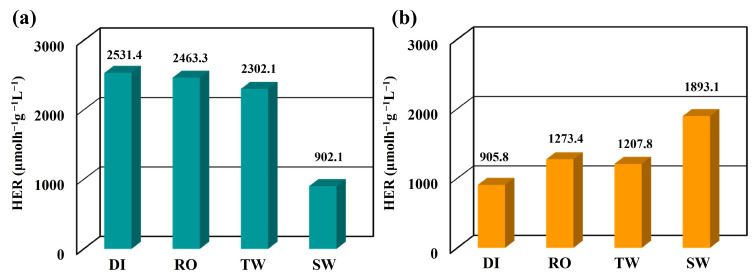
The average HER of Ag/Ag_2_S/CdS heterostructures for the different water sources with (**a**) the pH value adjusted to 9 and (**b**) an unadjusted pH value under visible light excitation.

**Figure 10 nanomaterials-13-01815-f010:**
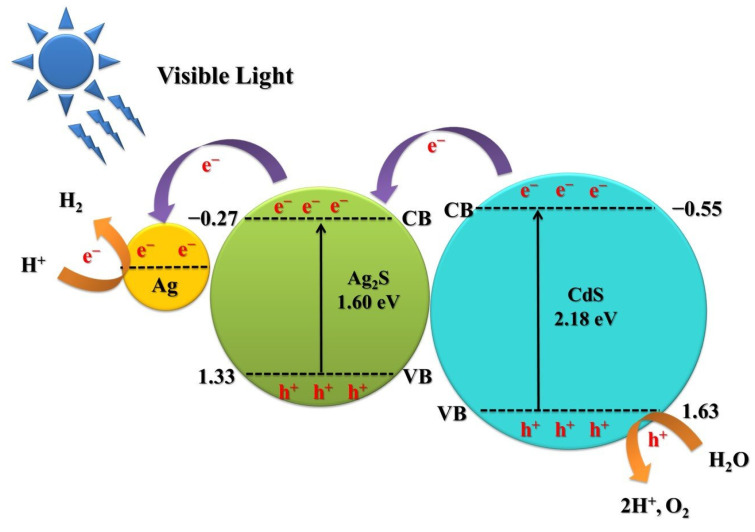
Proposed band structures and electron transfer in the Ag/Ag_2_S/CdS heterostructures.

## Data Availability

Not applicable.

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
