# Peer review of "Construction of Ag/Ag2S/CdS Heterostructures through a Facile Two-Step Wet Chemical Process for Efficient Photocatalytic Hydrogen Production"

_nanomaterials, 2023, doi:10.3390/nano13121815_

Round 1

Reviewer 1 Report

In this manuscript, the author has highlighted a two-step wet chemical approach for synthesizing ternary Ag/Ag2S/CdS heterostructures for efficient photocatalytic hydrogen evolution. However, the presented data do not fully support their conclusion. I recommend that this research paper be completely rejected.

English should be polished

Author Response

There are eight significant findings in the present study:

  1. The Ag/Ag2S/CdS heterostructures can be successfully synthesized via a facile two-step wet chemistry method.
  2. The concentration of CdS precursors and reaction temperature can be essential in inferencing photocatalytic water splitting under visible light excitation.
  3. The effect of operational parameters (such as the pH value, sacrificial reagents, reusability, water bases, and light sources) on the photocatalytic hydrogen production of Ag/Ag2S/CdS heterostructures was investigated.
  4. Ag/Ag2S/CdS heterostructures exhibited a 3.1-fold enhancement in photocatalytic activities compared to bare CdS nanoparticles under visible light excitation.
  5. Ag/Ag2S/CdS heterostructures can provide a higher light harvesting ability, and separating electron/hole pairs results in higher photocatalytic performance.
  6. Furthermore, the as-prepared Ag/Ag2S/CdS heterostructures possess excellent photocatalytic recyclability and stability over prolonged photocatalytic reactions.
  7. Furthermore, seawater's Ag/Ag2S/CdS heterostructures displayed approximately 2.09 times higher than in de-ionized water without adjusted pH values under visible light excitation.
  8. The ternary Ag/Ag2S/CdS heterostructures provide the new potential for designing efficient and stable photocatalysts for photocatalytic hydrogen evolution.

Reviewer 2 Report

In this manuscript ternary composites of Ag/Ag2S/CdS were synthesized and characterized with XRD, TEM and XPS. The structures were differed on the concentration of CdS and the reaction temperatures. These parameters were evaluated towards photocatalytic hydrogen evolution in water. It was shown that H2 production was affected by many aspects such as the concentration and the temperature preparation of CdS. The Ag/Ag2S/CdS heterostructures showed 3.1-fold higher H2 production activity compared to bare CdS. Moreover, it was determined that the pH and the type of water used during the photocatalytic reaction effect the H2 evolution rate. It was concluded that seawater produced 2.09 times higher H2 compared to de-ionized water under visible light irradiation.

The manuscript is well written, but needs some minor English editing. At the part where the synthesis of Ag/Ag2S/CdS is described the authors must add the specific reaction temperatures that they mention. The column used for GC used must be stated. A figure showing the curve of H2 production overtime must be included. The photocatalytic mechanism must be added at the end of the result and discussion part. Also, the H2 evolution rate of similar systems must be referred in order the readers to be able to compare the H2 evolution rate of Ag/Ag2S/CdS with analogous systems.

Minor English editing

Author Response

Response: Thanks for the pertinent and positive comments. We have amended the revised manuscript's descriptions (such as English and temperatures).

Subsequently, the hydrogen produced was quantified using gas chromatography (GC, Shimadzu GC-2014, Kyoto, Japan) equipped with a packed column (MS-5A, 60/80 mesh) and a thermal conductivity detector (TCD).

We have added the figure showing the curve of hydrogen production.

The photocatalytic mechanism has been added at the end of the result and discussion.

The Ag/Ag2S/CdS heterostructures exhibit higher HER values compared to other reported studies, including flower-like ZnO/Au/CdS nanorods [1], ZnO/ZnS/CdS heterostructures [2], Ag-based g-C3N4 composites [3], Cu2O/CuS/ZnS nanocomposite [4], and CdS/Pt-N-TiO2 nanocatalysts [5].

Reference:

  1. Li, Y.; Liu, T.; Feng, S.; Yang, W.; Zhu, Y.; Zhao, Y.; Liu, Z.; Yang, H.; Fu, W. Au/CdS Core-Shell Sensitized Actinomorphic Flower-Like ZnO Nanorods for Enhanced Photocatalytic Water Splitting Performance. Nanomater. 2021, 11, doi:10.3390/nano11010233.
  2. Guo, X.; Liu, X.; Yan, J.; Liu, S.F. Heterointerface Engineering of ZnO/CdS Heterostructures through ZnS Layers for Photocatalytic Water Splitting. Chem. Eur. J. 2022, 28, e202202662, doi:https://doi.org/10.1002/chem.202202662.
  3. Zhao, S.; Wu, J.; Xu, Y.; Wang, Z.; Han, Y.; Zhang, X. Ag2CO3-derived Ag/g-C3N4 composite with enhanced visible-light photocatalytic activity for hydrogen production from water splitting. Int. J. Hydrog. Energy 2020, 45, 20851-20858, doi:https://doi.org/10.1016/j.ijhydene.2020.05.191.
  4. Chang, Y.-C.; Chiao, Y.-C.; Fun, Y.-X. Cu2O/CuS/ZnS Nanocomposite Boosts Blue LED-Light-Driven Photocatalytic Hydrogen Evolution. Catalysts 2022, 12, doi:10.3390/catal12091035.
  5. Solakidou, M.; Giannakas, A.; Georgiou, Y.; Boukos, N.; Louloudi, M.; Deligiannakis, Y. Efficient photocatalytic water-splitting performance by ternary CdS/Pt-N-TiO2 and CdS/Pt-N,F-TiO2: Interplay between CdS photo corrosion and TiO2-dopping. Appl. Catal. B: Environ. 2019, 254, 194-205, doi:https://doi.org/10.1016/j.apcatb.2019.04.091.

Reviewer 3 Report

The manuscript «Construction of Ag/Ag 2 S/CdS heterostructures through a facilentwo-step wet chemical processes for efficient photocatalytic hydrogen production» represents study of two-step wet chemical approach for synthesizing ternary Ag/Ag 2 S/CdS heterostructures for efficient photocatalytic hydrogen evolution. Authors investigated the influence of preparation conditions (two-step wet chemical approach for synthesizing ternary Ag/Ag 2 S/CdS heterostructures for efficient photocatalytic hydrogen evolution) and operational parameters on the catalytic properties. XRD,FESEM, FETEM, XPS, PL, UV-Vis techniques were utilized. Although, it is an interesting study, there are several aspects that should be reviewed before it can be accepted for publication.

In Introduction, the purpose of article is absent. Therefore, there is no correlation between the topic of this study and actuality. Novelty of work should be improved.

 «The intensity of diffraction peaks for Ag decreases gradually as the reaction temperature increases. However, when the reaction temperature exceeds 180°C, CdS nanoparticles tend to self-aggregate instead of reacting with Ag nanowires to form Ag/Ag 2 S/CdS heterostructures [22]. This phenomenon leads to an increase in the intensity of the Ag diffraction peaks and a decrease in the Ag2S diffraction peaks.» In the work, the relative phase composition was analyzed by visual analysis of peak intensity. To improve the clarity of work, Rietveld refinement will help to estimate the quantitative phase analysis.

According to XRD data (figures 1 and 2), the peak of CdS reflections is wider than for Ag and AgS phases. The peak width is associate is with crystallite size. Is there a relationship between catalytic properties and crystallite sizes.

Author Response

Response: Thanks for the pertinent and positive comments.

Response: Thanks for your reminder. Moreover, most water used in photocatalytic water splitting for hydrogen generation is either de-ionized or pure water [1]. However, de-ionized and pure water requires freshwater purification, necessitating additional resources [2]. Limited research is available on the photocatalytic decomposition of seawater to produce hydrogen [3]. Considering the growing scarcity of freshwater resources, utilizing seawater directly for photocatalytic decomposition to generate hydrogen could help minimize freshwater consumption [4]. Thus, this study aims to produce hydrogen through the photocatalytic decomposition of seawater using a ternary Ag/Ag2S/CdS heterostructure.

Response: Thanks for your reminder. Moreover, the average sizes of Ag2S can be determined using Scherrer's formula based on X-ray line broadening: D = 0.9λ/βcosθ. Here, D represents the crystallite size, λ (equal to 1.54096 Å) denotes the X-ray wavelength, β represents the full width at half-maximum (FWHM), and θ refers to the diffraction angle. The average crystalline sizes of the Ag2S are calculated using the (−103) peak under the different reaction temperatures. The average crystalline sizes of Ag2S are 2.14 (120°C), 2.39 (140°C), 2.66 (160°C), and 1.86 nm (180°C), respectively. The crystalline sizes of Ag2S exhibit an opposite trend compared to the intensity of the Ag diffraction peaks. This finding confirms that at a reaction temperature of 180°C, CdS nanoparticles tend to self-aggregate instead of reacting with Ag nanowires to produce Ag/Ag2S/CdS heterostructures.

Reference:

  1. Lin, Y.-R.; Chang, Y.-C.; Chiao, Y.-C.; Ko, F.-H. Au@CdS Nanocomposites as a Visible-Light Photocatalyst for Hydrogen Generation from Tap Water. Catalysts 2023, 13, doi:10.3390/catal13010033.
  2. Bernauer, T.; Böhmelt, T. International conflict and cooperation over freshwater resources. Nat. Sustain. 2020, 3, 350-356, doi:10.1038/s41893-020-0479-8.
  3. Chang, Y.-C.; Zeng, C.-J.; Chen, C.-Y.; Tsay, C.-Y.; Lee, G.-J.; Wu, J.J. NiS/Pt loaded on electrospun TiO2 nanofiber with enhanced visible-light-driven photocatalytic hydrogen production. Mater. Res. Bull. 2023, 157, 112041, doi:https://doi.org/10.1016/j.materresbull.2022.112041.
  4. Wang, A.; Liang, H.; Chen, F.; Tian, X.; Yin, S.; Jing, S.; Tsiakaras, P. Facile synthesis of C3N4/NiIn2S4 heterostructure with novel solar steam evaporation efficiency and photocatalytic H2O2 production performance. Appl. Catal. B: Environ. 2022, 310, 121336, doi:https://doi.org/10.1016/j.apcatb.2022.121336.
